# Oxygen nanoclustering evades inverse Hall-Petch softening

Xiaolong Yu [1,2], Xilei Bian [1,2] ✉, Chang Liu [3] ✉, Qing Wang [1,2] ✉, Daniel Şopu [4], Daniel Kiener [5], Yifeng Li[6], Ge Wu [3], Yuan Wu [7], Yong Yang [8,9], Jürgen Eckert[4,5] & Gang Wang [1,2] ✉

Grain refinement can drastically increase the strength of metals. However, this approach tends to become less effective or even inverses once grain sizes are reduced to very small scales, generally below 10 ~ 20 nanometers. This softening effect emerges from grain boundary instability and the limited ability of dislocations to form and move within such nanosized grains. However, grain boundary stability can be tuned by solute decoration or grain boundary relaxation. In this context, we present a strategy to achieve superior strength and plasticity in nanograined metals simultaneously. The formation of oxygen (O)-rich clusters at grain boundaries can significantly improve grain boundary stability, even at the 3 nm grain size model $(CoCrNi)_{87}O_{13}$ (at.%) alloy investigated in this study. Furthermore, the presence of O-rich clusters in grain interiors promotes the accumulation and multiplication of dislocations, which facilitates strain hardening during deformation. Consequently, despite being situated in the inverse Hall-Petch regime, this $(CoCrNi)_{87}O_{13}$ alloy exhibits a remarkable yield strength of ~3.6 GPa and retains a uniform plastic strain of over 50% under micropillar compression. These findings therefore provide a universal design strategy for nanograined metals aimed at utilizing O clusters to achieve the highly desired combination of high strength and large deformability.

The mechanical behavior of polycrystalline materials is related to their grain size. When the grain size is reduced to the nanoscale, nanocrystalline alloys, due to the high density of grain boundaries (GBs), exhibit outstanding yield strength but extremely limited plasticity compared to their bulk micrometer-sized counterparts[1,2]. As grain size decreases, the nucleation stress for both perfect and partial dislocations increases[3], while the distance between GB sources and GB sinks of dislocations reduces[4]. This eventually results in a transition from dislocation- to GB-mediated plastic deformation (GB diffusion creep and/or GB sliding)[5–7]. Meanwhile, due to the higher energy[8] and fewer bonds at GBs[9], the deformation behavior mediated by GBs often results in localization and softening, facilitating the nucleation and propagation of shear bands and cracks at the GBs[10,11].

To suppress GB-dominated deformation and promote dislocation-mediated processes, strategies such as high-pressure techniques[12], nanotwining[2], and composition undulation (inter and intra granular)[13–16]

[1]State Key Laboratory of Materials for Advanced Nuclear Energy, Shanghai University, Shanghai 200444, China. [2]Zhejiang Institute of Advanced Materials, Shanghai University, Jiashan 314100, China. [3]State Key Laboratory for Mechanical Behavior of Materials, Xi'an Jiaotong University, Xi'an 710049, China. [4]Erich Schmid Institute of Materials Science, Austrian Academy of Sciences, Leoben 8700, Austria. [5]Department Materials Science, Montanuniversität Leoben, Leoben 8700, Austria. [6]Laboratory for Microstructures, Shanghai University, Shanghai 200444, China. [7]State Key Laboratory for Advanced Metals and Materials, University of Science and Technology Beijing, Beijing 100083, China. [8]Department of Mechanical Engineering, College of Engineering, City University of Hong Kong, Hong Kong 999077, China. [9]Department of Materials Science and Engineering, College of Engineering, City University of Hong Kong, Hong Kong 999077, China. ✉e-mail: bianxilei@shu.edu.cn; chang.liu@xjtu.edu.cn; qingwang@shu.edu.cn; g.wang@shu.edu.cn

are employed to enhance GB stability and facilitate dislocation nucleation. Among these strategies, composition undulation offers greater versatility and simplicity in alloy design. It allows targeted incorporation of appropriate substitutional or interstitial atoms during the composition design process to introduce desired chemical inhomogeneities. Compared to substitutional atoms, interstitial atoms such as B, C, N, and O often have a more significant impact on enhancing the mechanical properties of nanocrystalline alloys[17–21]. This is because the introduction of interstitial atoms can cause stronger segregation tendency[9], greater affinity difference with different matrix elements[17], and larger lattice distortion[22]. However, that often poses challenges, such as the risk of forming brittle secondary phases (e.g. oxides, borides, nitrides), due to the excessive segregation at GBs, thereby limiting ductility.

The recently developed multi-principal element alloys (MPEAs) offer a potential solution to the above-mentioned issues. The interactions among multiple constituent elements result in ubiquitous localized lattice distortions[23] and sluggish diffusion[24] in MPEAs, making them suitable for accommodating large amounts of interstitials[21,25] with a reduced risk for forming secondary phases[26]. When the grain size of nanograined (NG) metals is reduced to below 10 nm, dislocation-mediated plastic deformation is accommodated by partial rather than perfect dislocations[3,14]. This suggests that the stacking fault energy (SFE), which determines the nucleation barrier for the formation of partial dislocations, has a pronounced effect on the plasticity of NG metals[27]. Therefore, to stabilize GBs and promote dislocation accumulation, we selected a CoCrNi MPEA with low SFE ($22 \pm 4$ mJ·m$^{-2}$)[28] as the matrix for solutioning with oxygen interstitials. Oxygen was selected based on its high solubility[25] and pronounced strengthening effect[17] in MPEAs. Additionally, oxygen tends to segregate into octahedral sites with a high local concentration of Cr or low local concentration of Ni[29], which contributes to enhancing the chemical inhomogeneity in the CoCrNi alloy.

In this work, magnetron sputtered nanograined CoCrNi alloys show inverse Hall-Petch softening. By introducing interstitial oxygen atoms, this softening effect can be effectively suppressed. Specifically, a supra-nano-grained (3 nm average grain size) model $(CoCrNi)_{87}O_{13}$ (at.%) alloy exhibits an ultra-high yield strength of ~3.6 GPa and a uniform compressive plastic strain of over 50%. The elemental distribution, chemical inhomogeneity, and structural changes in the samples before and after compression are revealed by three-dimensional atom probe tomography (3D-APT) and transmission electron microscopy (TEM). The underlying deformation mechanisms, especially the impact of oxygen nanoclustering dynamics on the superior mechanical properties, are disclosed. The current findings not only advance the development of high-performance nanostructured materials, but also offer perspectives and approaches for future material design.

## Results

### Structure and chemical analysis of the as-deposited CoCrNi-O MPEAs

Atom probe tomography (APT) was employed to investigate the nanoscale chemical inhomogeneity in the as-deposited CoCrNi-O alloys. Results indicate that this alloy has an average composition of $Co_{30.56}Cr_{28.74}Ni_{28.20}O_{12.50}$ (at.%). Hereafter referred to as $(CoCrNi)_{87}O_{13}$ (O-13) alloy, which is taken as the model material (Fig. 1a). To elucidate how oxygen content influences the structure and mechanical properties, we systematically modulated the oxygen content in CoCrNi alloys. The specific compositions of $(CoCrNi)_{95}O_5$ (O-5) and $(CoCrNi)_{70}O_{30}$ (O-30) alloys, with oxygen content of 4.81 at.% and 30.25 at.%, are shown in Supplementary Figs. 1 and 2, respectively. Based on the Co-O, Cr-O, and Ni-O binary phase diagrams, oxygen is expected to be present as supersaturated solid solution in the CoCrNi matrix (Supplementary Fig. 3). Significant chemical inhomogeneity at the nano-scale, resulting from the introduction of substantial amounts of oxygen atoms, was

readily detectable. Fig. 1b displays the spatial variation of the Cr concentration throughout the needle-shaped sample. An elemental concentration profile plotted parallel to the growth direction (GD) reveals that Cr and O elements exhibit a correlated variation trend, i.e., their concentrations simultaneously increase or decrease within ±2.50 at.% of the nominal composition (Fig. 1c). These nanoscale (Cr, O)-rich and (Co, Ni)-rich clusters are even more obvious in two-dimensional in-plane concentration maps (Supplementary Fig. 4). Similar composition undulations are also observed in O-5 and O-30 alloys (Supplementary Figs. 1 and 2).

To further investigate the unique nanostructure and chemical inhomogeneity caused by the high content of oxygen atoms, a spherical aberration-corrected transmission electron microscope (Cs-corrected TEM) equipped with an electron energy loss spectroscopy (EELS) detector was employed to simultaneously collect structural and elemental information at the nanoscale in the same region of the sample. Bright-field scanning transmission electron microscopy (BF-STEM) images reveal a supra-nano columnar-grained structure of the as-deposited O-13 MPEA (Fig. 1d) with an average grain size of ~3 nm (Fig. 1e). In this work, pure CoCrNi (O-free), O-5, and O-30 MPEAs also exhibit a similar columnar-grained structure with grain size spanning from 2.09 nm to 16.55 nm (Supplementary Fig. 5). All reported grain sizes in this study are defined by the transverse width of the columnar grains[25,30]. The average grain size for each sample was derived from the dark-field TEM (DF-TEM) images with more than 300 grains counted (Supplementary Fig. 6). Due to the low SFE of the CoCrNi MPEA[28,31,32], the CoCrNi and CoCrNi-O MPEAs have FCC or FCC + HCP dual-phase structure in absence of any oxides, as identified from the selected area electron diffraction (SAED) patterns (Fig. 1f) and the diffraction profiles (Supplementary Fig. 7) deduced from SAED patterns using PASAD-tools[33]. In addition, the SAED patterns reveal a slight (111) texture in the films, indicating a [111] preferential growth direction, which is a common feature in as-deposited films with FCC structure[21,34]. The high-angle annular dark-field (HAADF) image in Fig. 1g reveals nano-scale periodic composition fluctuations, which are not directly associated with GBs, but are uniformly distributed throughout the entire O-13 sample. The contrast in the STEM-HAADF image depends on the atomic number, whereby the bright areas correspond to regions enriched in heavy atoms[35]. Notably, the wavelength $\lambda$ of the composition fluctuations obtained from STEM-HAADF images is in the range of ~2.80 nm (transversal) to ~5.45 nm (vertical), which is close to the grain size (Fig. 1h).

Note that even individual nanograins contain compositional fluctuations (Fig. 1i and k). Besides, significant storage of partial dislocations inside the nanograins is evidenced, including $1/6 < 112>$ Shockley partial dislocations and $1/3 < 111>$ Frank partial dislocations. Furthermore, a high density of stacking faults (SFs) and nanotwins (NTs) caused by these partial dislocations can be identified. Most of them are parallel to the (111) texture and propagate across the entire grain (Fig. 1i). Typical columnar grains containing high densities of planar defects (SFs and NTs) and HCP phases are shown in Supplementary Fig. 8. Fig. 1j shows a high-angle GB (with a misorientation angle of 23°), implying that the region between nanograins is not amorphous, consistent with the diffraction patterns shown in Fig. 1e. Fig. 1k shows that there is no obvious structural difference between domains with different compositions, but a slight difference in lattice constants. The interplanar spacings of {111} crystal planes were statistically analyzed, showing values of ~2.07 Å in oxygen-enriched regions and ~2.02 Å in oxygen-depleted regions, i.e., an expansion of 1.0% and a contraction of 1.5%, respectively, compared to the average {111} interplanar spacing of 2.05 Å. Altogether, it is the unique structure of the CoCrNi-O MPEA characterized by a large volume fraction of GBs and a high density of SFs and NTs in grain interiors, that provides sufficient accommodation sites for the massive amount of interstitial oxygen atoms.

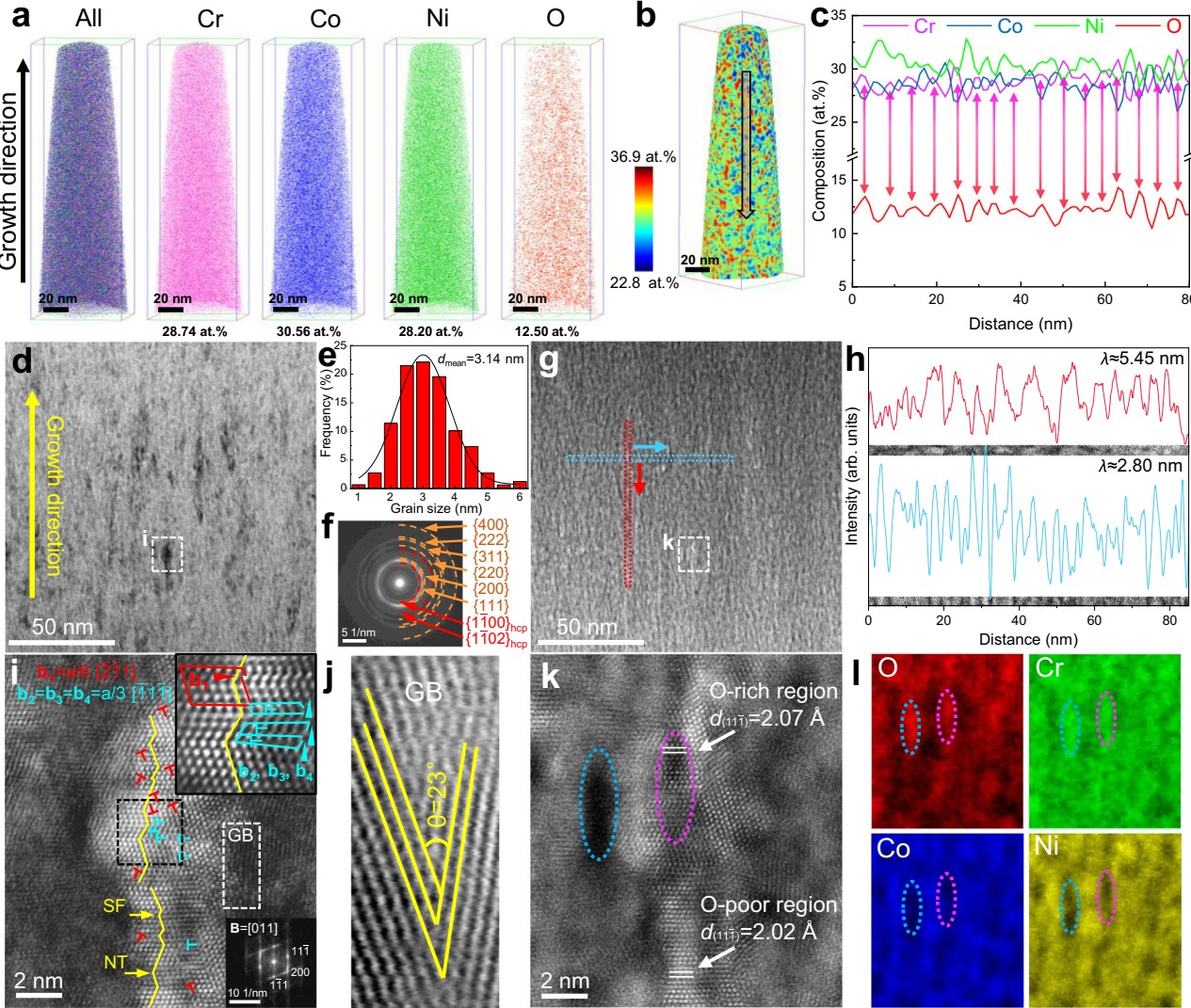

**Fig. 1 | Material microstructure with nanoscale chemical inhomogeneity.**
**a** Combined and individual 3D elemental maps of as-deposited O-13 MPEA with chemical composition in atomic percent (at.%) based on APT analysis. **b** 3D reconstruction map based on APT measurement, illustrating the spatial variation of Cr concentration throughout the sample. **c** 1D compositional profile along the length direction of the arrow displayed in (**b**). **d** Typical BF-STEM image showing the columnar nanocrystal structure. **e** Grain size (columnar width) distribution of the columnar nanograins. **f** The SAED pattern acquired from **d** shows nanocrystalline diffraction rings, corresponding to a mixture of HCP and FCC structures. **g** Typical STEM-HAADF image taken from the same region as **d**. **h** Intensity line profiles of the red and blue square region in (**g**). **i** DF-STEM image of the white-dashed square region in **d**. The red and blue "⊥" represent Shockley and Frank partial dislocation, respectively. The inset at the bottom right is the corresponding fast Fourier transform (FFT) image. The inset at the top right is the magnified image of the region corresponding to the black square, showing a Shockley partial dislocation and two adjacent Frank partial dislocations. **j** Enlarged high-angle GB image corresponding to the white-framed region in (**i**). **k** STEM-HAADF image captured from the same region as **i**. The oxygen-enriched ((Cr, O)-rich) and oxygen-depleted ((Co, Ni)-rich) regions exhibit different interplanar spacings along {111} crystal planes. **l** EELS mapping collected in the corresponding region of **k**. The blue and pink dashed frames are (Cr, O)-rich regions in the grain exterior and interior, respectively. Source data are provided as a Source Data file.

The specific elemental distribution of the identical region in Fig. 1k was determined by EELS, as shown in Fig. 1l. The bright regions in Fig. 1k are enriched in Co and Ni, while the dark regions are enriched in O and Cr. Furthermore, (Cr, O)-rich clusters can be observed both inside and outside the grains. This chemical inhomogeneity is ascribed to the different affinity of oxygen atoms to the other three elements. According to the Pauling scale or Allen scale[36], the electronegativity difference between O and Cr is much larger than between O and Co or Ni, respectively, suggesting that the chemical affinity is also larger between O and Cr (Supplementary Table 1). In the CoCrNi MPEA, the inherent chemical short-range order, specifically, atomic-scale enrichment of Cr regions[37,38], creates a chemical potential gradient for oxygen diffusion, which allows oxygen atoms to overcome

concentration gradients during the nucleation stage of film deposition, leading to sustained uphill diffusion. As a result, the oxygen content in the Cr-rich region increases continuously, causing the gradual growth of (Cr, O)-rich domains and, ultimately, creating nanoscale chemical inhomogeneity upon deposition.

The bonding of O with the metallic elements in the O-13 alloy was investigated by X-ray photoelectron spectroscopy (XPS) (Fig. 2), with an ~100 nm-thick top surface layer being removed by Ar⁺ etching. Most metallic atoms show zero oxidation states, while a small proportion shows positive oxidation states[39]. This result indicates that some metallic atoms got their electrons extracted by oxygen atoms, resulting in additional bonding contributions due to the presence of oxygen interstitials[40]. Given that the average concentration of O is 12.50 at.%,

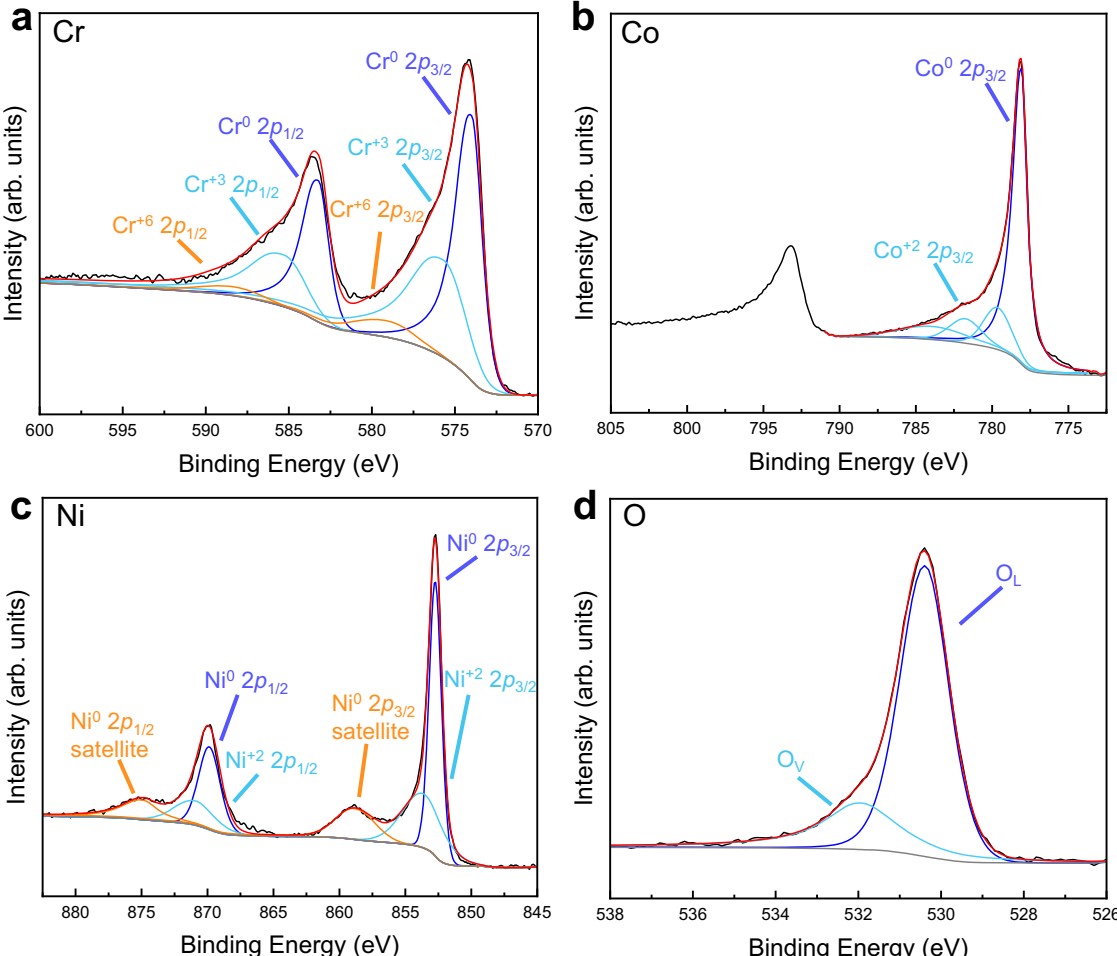

**Fig. 2 | Electronic characterizations of as-deposited O-13 MPEAs.** High-resolution XPS spectra of (**a**) Cr 2*p*, **b** Co 2*p*, **c** Ni 2*p*, and (**d**) O 1*s*, with deconvoluted peaks labeled with their corresponding chemical states. The $O_L$ and $O_V$ in (**d**) represent lattice oxygen and oxygen vacancy, respectively. The black, gray and red lines represent the raw spectrum, background, and the overall fitting curve, respectively. Source data are provided as a Source Data file.

the additional bonding contributions from O interstitials have a significant impact on the resistance against dislocation motion and GB sliding. In addition, a comparison of the 2*p* XPS spectra for Cr, Co and Ni reveals that Cr exhibits a higher proportion of oxidation states, which is consistent with the prior analysis of (Cr, O)-rich clusters. The above results are further substantiated by the EELS data collected from the region in Fig. 1k using the white line method (Supplementary Fig. 9). For oxygen, the XPS spectrum displays two peaks: $O_L$ (lattice oxygen) at ~530.4 eV and $O_V$ (oxygen vacancy) at ~531.9 eV[41]. For reference, the XPS data of the film surface and the pre-alloyed target (etching depth ~100 nm) were also collected (Supplementary Fig. 10).

**Mechanical properties**

The mechanical properties of the CoCrNi and CoCrNi-O MPEAs were investigated by nanoindentation tests. Fig. 3 displays the plot of hardness(*H*) versus grain size (*d*) or twin thickness (*t*) for Ni[42–45], NT-Ni[46], NiMo[13], NiCo[47], and CoCrNi[48–57] alloys with FCC structure. The hardness of all these alloys continuously increases with the decreasing grain size over the range of larger than ~10-20 nm, which is quantitatively consistent with the Hall-Petch relation[58,59]. However, below this threshold value, NiMo, NiCo and CoCrNi alloys show a breakdown of Hall-Petch relations, resulting in an inverse Hall-Petch relation, i.e., further reducing grain size leads to a decrease in strength. In stark contrast, the CoCrNi-O alloy with varying oxygen content (4.81–30.25 at.%) in the present work, which is composed of extremely

small grains of ~3-8 nm and contains a large content of interstitial oxygen atoms, however, remarkably evades the inverse Hall-Petch trend of CoCrNi alloys, thus retaining an ultra-high hardness of over 11.30 GPa. Moreover, it should be noted that continuous strengthening of Ni can also be achieved with NT-Ni[46]. However, this phenomenon of strengthening in NT-Ni is accompanied by a dramatical decrease in the plastic deformability, which is significantly distinct from our CoCrNi-O alloy with large uniform deformation.

To shed light on the thermally activated mechanisms contributing to plastic deformation, the strain rate sensitivity and activation volume of the CoCrNi-O MPEA were obtained through nanoindentation tests with varying strain rate[60,61] Taking O-13 alloy as an example, Supplementary Fig. 11 provides a detailed presentation based on the nanoindentation results. As shown in Supplementary Table 3, all investigated CoCrNi-O MPEAs exhibit a positive strain-rate sensitivity ~0.02. Accordingly. the activation volume is estimated to be ~8.43×$10^{-29}$-9.63×$10^{-29}$ $m^3$, which is equal to ~29-33$b^3$ (b = a/6 | < 211 >| assuming partial dislocation to govern plasticity). Based on the theory of Taylor[62], the area swept by the dislocation during activation is 0.60-0.68 $nm^2$ (29-33$b^2$), which are much smaller than the grain area (>9 $nm^2$). Moreover, the magnitude of the activation volume is quite close to that of most metals with ultrafine grain size (~100–1000 nm), indicating a regime where plastic deformation is mediated by partial dislocations[60]. To characterize the strength and plasticity of the CoCrNi-O MPEA, room-temperature uniaxial micropillar (500 nm and 1 µm in diameter)

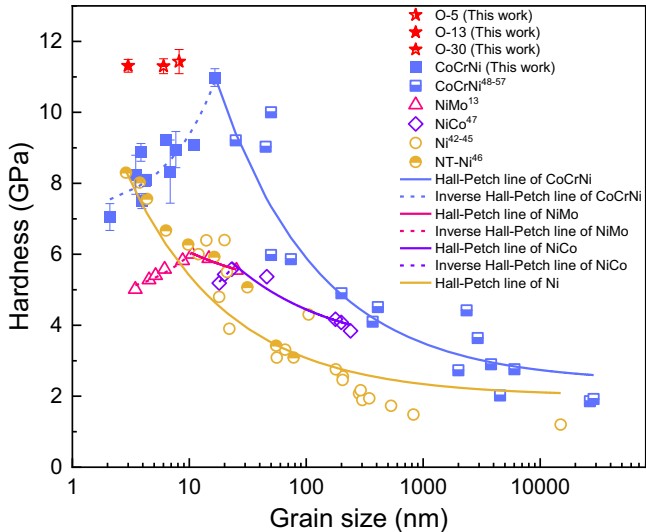

**Fig. 3 | Evading inverse Hall-Petch softening in the CoCrNi alloy.** Hardness versus average grain size for our current CoCrNi and CoCrNi-O alloys. Error bars represent standard deviation. Literature data of Ni[42–45], NT-Ni[46], NiMo[13], NiCo[47], and CoCrNi[48–57] alloys are included for comparison. The solid lines are fitted by Hall-Petch relation ($H$-$d^{-\frac{1}{2}}$). The dash lines show the inverse Hall-Petch softening ($H$-$d$). The fitting parameters for the Hall-Petch and inverse Hall-Petch relationships are summarized in Supplementary Table 2. Source data are provided as a Source Data file.

compression tests were conducted under displacement-controlled mode at a constant strain rate of $4 \times 10^{-3}\,\text{s}^{-1}$. Each compression test was repeated three times. The Supplementary Fig. 12 presents the morphology of the as-fabricated O-13 micropillars. Representative compressive engineering stress-strain curves are shown in Fig. 4a. The yield strength ($\sigma_y$) of O-13 micropillars with a diameter of 500 nm is $3.62 \pm 0.21$ GPa, which is comparable to that of micropillars with a diameter of 1 μm ($3.60 \pm 0.10$ GPa), showing a marginal size effect. This is because the diameters of the micropillars, i.e., 500 nm or 1 μm, are much larger than the grain size (3 nm), and thus the effect of the extrinsic size of the micropillars on the mechanical properties is almost negligible. In stark contrast to the O-free CoCrNi alloy with a grain size of ~3.93 nm, our 3 nm grain size O-13 alloy is much stronger, with more than 1.00 GPa higher yield strength than found for micropillars with 500 nm diameter. In addition, the yield strength of O-5 and O-30 micropillars with a diameter of 500 nm are $3.35 \pm 0.08$ GPa and $4.14 \pm 0.12$ GPa, respectively. It seems that the yield strength increases with the oxygen content. More intriguingly, our O-13 alloy exhibits stable plastic flow up to a compressive strain of 75%, which is twice higher than for the O-free CoCrNi nanocrystals. Moreover, among the current investigated three O-containing CoCrNi alloys, the O-13 alloy has the smallest grain size and exhibits an optimal combination of strength and plasticity. The serrated-like plastic flow may result from dislocation motion and activation of slip systems during plastic deformation[63]. It is also evident from the stress-strain curves that the studied alloy sustains strain-hardening ability without catastrophic load drops, in stark contrast to the findings reported for CoCrNi micropillars[29,63]. This stable plastic deformation under large strain is of importance for nanocrystalline materials. To substantiate the presence of strain hardening, three 500-nm-diameter micropillars, extracted from 1-μm-diameter micropillars deformed to 30% using focused ion beam (FIB), were tested under identical conditions. The extracted micropillars show a yield strength of $4.16 \pm 0.10$ GPa, which is approximately 500 MPa higher than that of the micropillars prepared from the undeformed material. Morphological observations of the deformed micropillars further demonstrate that all O-13 micropillars

maintain high deformability without the localization of shear deformation (Fig. 4b–d). Similar delocalization morphology is also observed in the O-5 micropillars, while the O-30 and O-free ($d$ ~ 3.93 nm) micropillars show obvious shear deformation (Supplementary Fig. 13). Compared to other FCC structured alloy systems, including coarse-grained (CG) or NG CoCrNi MPEAs[32,34,48,49,53,63–70], pure Ni metal[46], AlFeCoNiC$_x$ ($x = 0$, 0.5, 1.0, 2.0, 4.1) MPEAs[71], and Al$_{0.1}$CoCrFeNi MPEAs[72] reported previously, with a very high yield strength of over 3.60 GPa and an exceptional homogeneous plastic deformability, our CoCrNi-O alloy situates at the upper-right corner of a respective property map (Fig. 4e), impressively overcoming the strength-plasticity trade-off and the brittleness associated with extreme grain refinement. This can be attributed to the unusually rugged potential energy landscape created by the interstitial oxygen-induced chemical inhomogeneity for dislocation motion and GB sliding[13,14]. A detailed analysis of this aspect will be presented in the discussion section.

Notably, our oxygen nanoclustering strategy extends beyond CoCrNi to diverse alloy systems, as demonstrated in nanocrystalline FeCrNi-O (6.86 at.% O) and amorphous HfNbTiZr-O (15.00 at.% O) alloys. For FeCrNi-O, oxygen-induced chemical heterogeneity elevates hardness to 9.00 GPa (yield strength ~2.98 GPa) while sustaining >50% uniform strain (Supplementary Fig. 14). This represents a 15% strength enhancement over oxygen-free nanocrystalline FeCrNi[73], effectively overcoming the strength-plasticity trade-off. Similarly, amorphous HfNbTiZr-O achieves a remarkably high yield strength of 4.20 GPa (Supplementary Fig. 15), which is more than twice the literature benchmarks[63]. These results establish oxygen nanoclustering as a universal paradigm for designing ultra-strong nanostructured materials with exceptional plasticity. We anticipate these strong and deformable thin films have widespread technological applications in extreme mechanical environments requiring supreme wear resistance such as next-generation cutting tools and protective coatings.

## Evolution of microstructure and chemical inhomogeneity during deformation

To investigate the impact of the high concentration of interstitial oxygen atoms and the associated chemical inhomogeneity on the deformation behavior of CoCrNi-O micropillars, TEM and APT were employed for detailed structural and compositional characterization of the deformed micropillars (O-13 as an example). Fig. 5a depicts a cross-sectional STEM-HAADF image of a micropillar deformed to 50% strain, revealing that the grain morphology has been changed from columnar to globular, with an average size of ~2.79 nm. This is due to the high initial density of partial dislocations which interact with the columnar GBs under stress, leading to the formation of new subgrain boundaries and a reduction in grain size along the GD (i.e., the compression axis). To further elucidate the changes in grain morphology under compressive stress, longitudinal cross-sectional DF-TEM images of micropillars at 0%, 30%, and 50% strain were obtained (Supplementary Fig. 16). The change in grain morphology due to dislocation movement is observable in the 30% strained sample. Note that the size of the globular grains does not increase with increasing strain, confirming the absence of stress-assisted grain growth. This suggests that the GBs in the O-13 MPEA remain stable upon deformation. Fig. 5b and c reveal that even after 50% deformation, the micropillar contains a number of partial dislocations and twins within individual grains. This indicates that partial dislocations constantly accommodate plastic strain during deformation. EELS spectra acquired from the region of Fig. 5a and an elemental line scan across a grain were performed to investigate elemental redistribution during deformation (Fig. 5d). Despite deformation, chemical inhomogeneity persists within the O-13 MPEA with (Cr, O)-rich clusters becoming more concentrated at the GB regions, as compared to the pre-deformed sample (Supplementary Fig. 17). This is because oxygen atoms are easily dragged by moving dislocations, and GBs often serve as traps for dislocations[4,40].

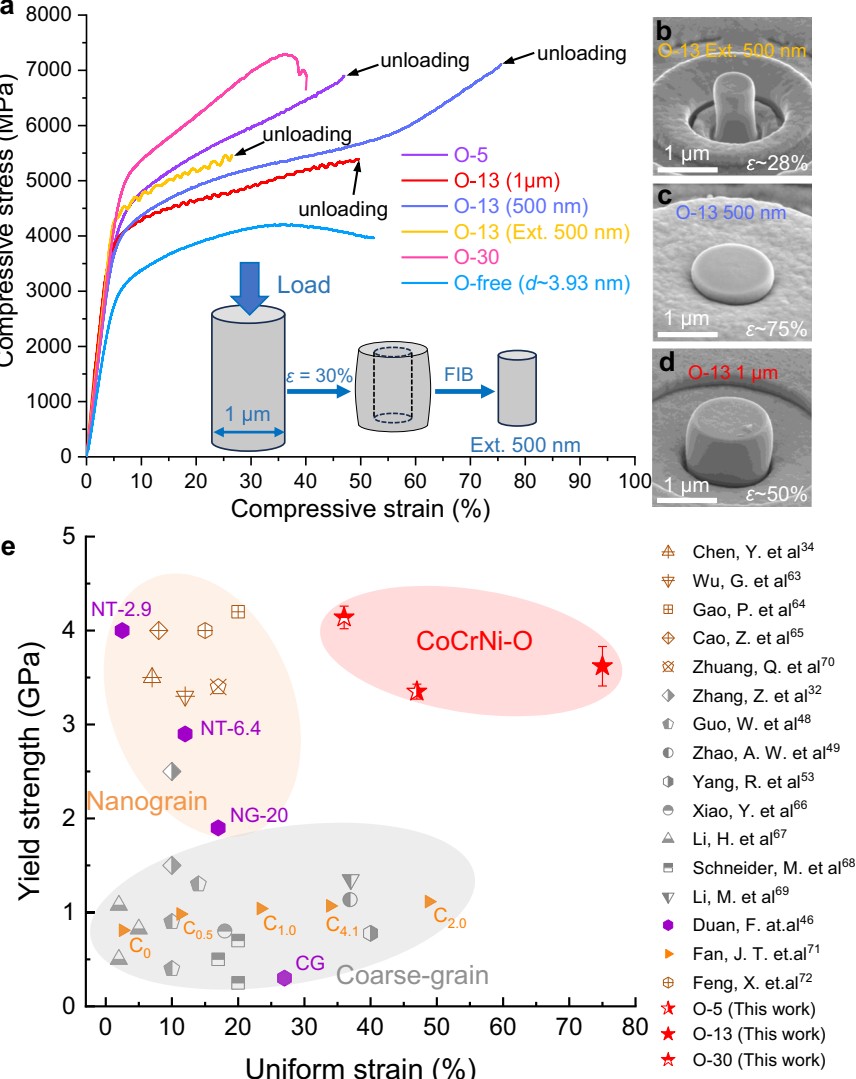

**Fig. 4 | Mechanical properties of the CoCrNi-O MPEAs. a** Compressive engineering stress-strain curves of the micropillars with diameters of 500 nm and 1 μm of the CoCrNi-O alloys tested with identical conditions at room temperature. Insert shows how the Ext. 500 (extracted 500 nm) micropillars were obtained. **b–d** SEM images of the tested O-13 micropillars after compression. No shear bands can be observed. **e** Yield strength *vs.* uniform strain of the CoCrNi-O alloys, in comparison with other FCC structured alloy systems, including O-free CoCrNi alloy[32,34,48,49,53,63–70], pure Ni metal with grain size of 80 μm (CG) and 20 nm (NG-20) or twin thickness of 6.4 nm (NT-6.4) and 2.9 nm (NT-2.9)[46], AlFeCoNiC$_x$ ($x$ = 0, 0.5, 1.0, 2.0, 4.1) alloy[71], and Al$_{0.1}$CoCrFeNi alloy[72] tested in micropillar compression at ambient temperature. Error bars represent standard deviation. The data of this work are located at the upper-right corner. Source data are provided as a Source Data file.

Furthermore, the attraction for oxygen to GBs is greater than that within the grains[9], leading to a continuous migration of oxygen from the grain interior to the GB regions.

In the APT results, we highlight the O-rich region using 16 at.% O-isoconcentration surfaces (named as 16%-O). After deformation, the volume fraction of this iso-surface increases from ~0.21 vol.% to ~0.51 vol.%. In a 30 × 30 × 100 nm³ cubic region (Fig. 5e), the number of 16%-O increases from 54 before deformation to 84 after deformation. We also statistically analyzed the average volume of 16%-O before and after deformation, which on average increases from ~4.02 nm³ to ~7.17 nm³ (Fig. 5f). Additionally, statistics on the nearest-neighbor distance of oxygen shows that after deformation, the nearest-neighbor distance decreases from ~6.35 nm to ~5.85 nm (Fig. 5g). These observations suggest that deformation does not lead to elemental homogenization, but instead exacerbates chemical inhomogeneity, especially for the oxygen distribution. Radial distribution function (RDF) analysis was performed on the APT dataset obtained from the samples before and

after deformation (Fig. 5h and Supplementary Fig. 18) by analyzing the average local neighborhood around a reference type of atom as a function of radial distance. Fig. 5h shows that after deformation, O-O interactions become more positive at distances below approximately 3 nm. Moreover, the bulk normalized concentration of O around the central O atom decreases to a minimum value below 1 and then increases with increasing distance, exceeding 1 and exhibiting a distinctive core-shell concentration distribution. Similar trends are also observed in the RDFs calculated for Cr, Co, and Ni as central atoms (Supplementary Fig. 18).

## Discussion

The observation of a high density of partial dislocations within grains in both pre- and post-deformation O-13 samples indicates that, despite the reduction in grain size to ~3 nm, dislocations remain a main carrier of plastic deformation in the O-13 MPEA rather than primarily GBs. This can be rationalized by the fact that the low strain rate sensitivity (0.02)

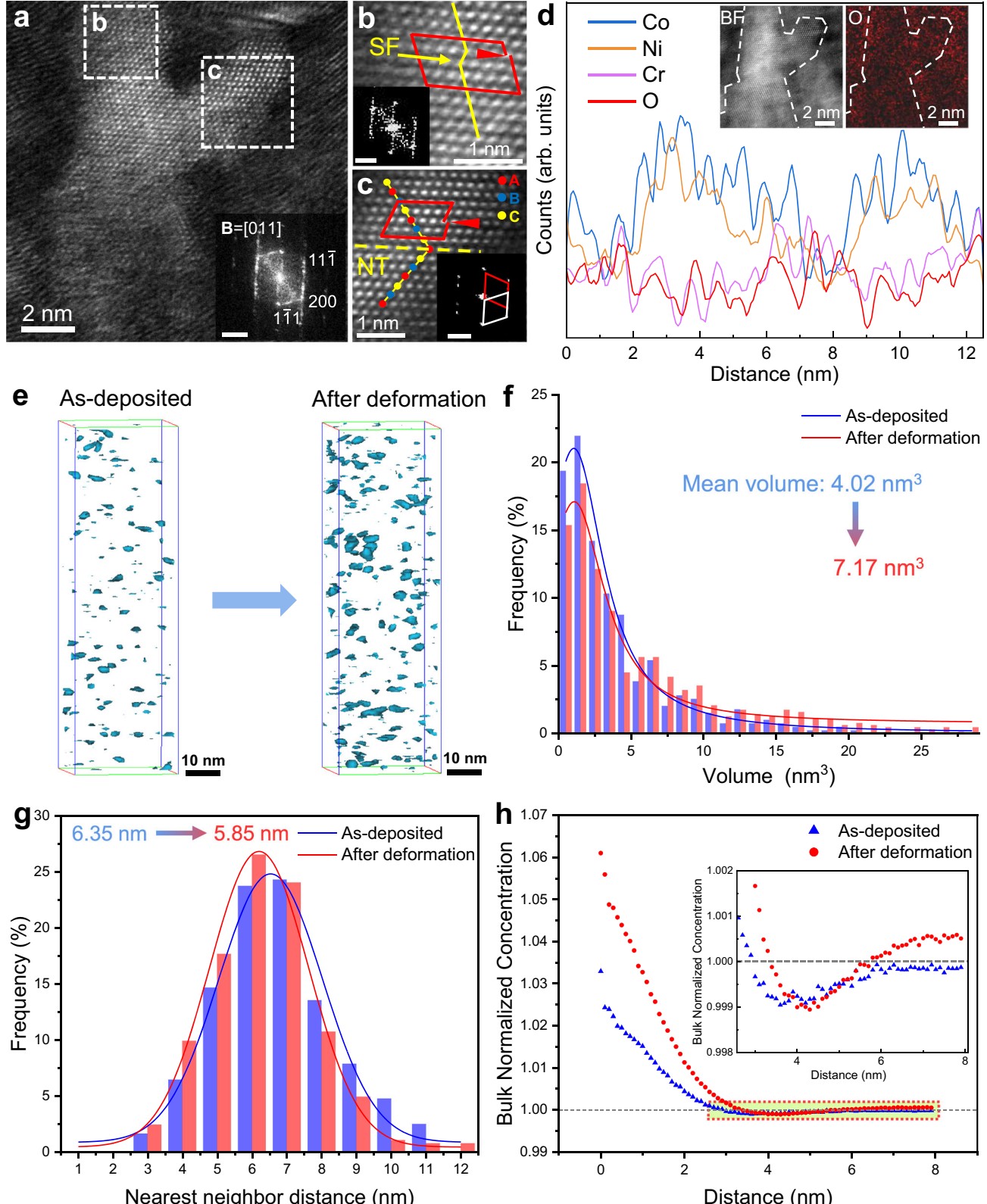

**Fig. 5 | Deformation mechanisms of the CoCrNi-O MPEAs. a** Typical DF-STEM image of the O-13 MPEA upon 50% compressive strain. The inset image is the corresponding FFT pattern. Scale bar of inset is 5 1/nm. **b, c** Magnified images of the white dashed square regions in (**a**), showing a SF with a Shockley partial dislocation and a coherent twin boundary with a Shockley partial dislocation, respectively, as verified by their corresponding FFT patterns (insets). Scale bars of insets are 10 1/nm. **d** STEM-EELS line-scanning analysis. The insets show the BF-STEM image and EELS mapping of oxygen corresponding to the acquisition area. The white dashed lines denote grain agglomerates and the yellow arrow represents the line-scanning direction. **e–h** APT characterization of the structure evolution before and after compression. **e** Changes in spatial distribution of 16 at.% O-isoconcentration surfaces (16%-O). **f** 16%-O volume distribution reveals that the size of O-rich cluster becomes larger. **g** Nearest neighbor distance distribution for 16%-O. **h** Variation of calculated O-O RDFs. Source data are provided as a Source Data file.

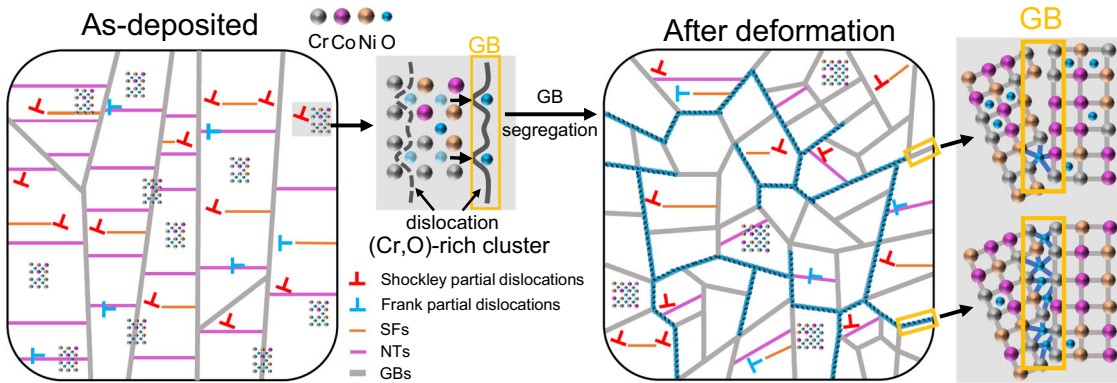

**Fig. 6 | Schematic diagram illustrating the oxygen redistribution dynamics in the CoCrNi-O MPEAs.** Upon plastic straining, Shockley partial dislocations in the alloy can slide along close-packed planes, while Frank partial dislocations can climb along coherent twin boundaries to accommodate the plastic deformation, resulting in the columnar grains transformed into globular grains. The SFs and NTs induced by partial dislocations, along with the (Cr, O)-rich clusters induced by oxygen in grain interiors, serve to hinder dislocation motion and promote accumulation and multiplication of dislocations, which enhance strain hardening during deformation. In the interaction between partial dislocations and the (Cr, O)-rich clusters, oxygen atoms are dragged to the GBs and gradually accumulate at these GBs, thereby increase the O-M bonds at the GBs, increasing the GB strength and stabilizing them.

and high activation volume ($32b^3$), which are different from the typical values of strain rate sensitivity (-0.05) and activation volume (smaller than $10b^3$) that mediated by GBs[60], implying GB activation is effectively suppressed in this O-13 alloy with grain size of 3 nm. The high density of intra-granular partial dislocations in the samples before and after deformation is primarily attributed to the extremely low SFE (-22 mJ·m$^{-2}$) of the CoCrNi MPEA[28]. According to the dislocation-based model proposed by Chen et al.[3], there is a theoretical critical grain size ($d_c$) below which the critical nucleation stress of Shockley partial dislocations will be less than that of full dislocations. Taking a SFE of 22 mJ·m$^{-2}$, the critical grain size of O-13 alloy is calculated to be approximately 87.94 nm (see Methods), which is significantly larger than the grain size of 3 nm. If the SFE of CoCrNi is in fact negative as reported in theoretical calculations[31,32], the formation of partial dislocations or SFs becomes an energy-reduction process. Therefore, the CoCrNi-O MPEAs in this study can easily overcome the energy barrier and continuously form partial dislocations under the action of external loads. Although the addition of interstitial oxygen atoms may increase the SFE of the alloy[74], previous reports indicate that a large number of partial dislocations and twins can still be found in CoCrNi-O alloys, even upon addition of 6 at.% oxygen[75]. In addition, according to the model proposed by Nieh and Wadsworth[76], the critical size $l_C$ of a grain to accommodate a single Shockley partial dislocation for the O-13 MPEA was estimated to be 1.14 nm (see Methods), indicating that even after deformation, individual nanograins in the CoCrNi-O alloy can still accommodate a certain number of partial dislocations. Meanwhile, the high density of GBs in the CoCrNi-O alloy can also provide a large number of nucleation sites for partial dislocations[3,4].

The excellent dislocation storage capability can be attributed to the large lattice friction stress and the chemical inhomogeneity induced by the high concentration of interstitial oxygen atoms. On one hand, the massive incorporation of interstitial oxygen atoms can effectively trap dislocations and hinder their motion. On the other hand, the intra-granular chemical inhomogeneity can present a roughened potential energy landscape for moving dislocations and force them to retard and modify their configuration[14]. The low SFE decreases the nucleation stress for partial dislocations, and the obstructions for dislocation motion reduce the mean free path of partial dislocations. Meanwhile, the instability of GBs can be effectively mitigated by the enhanced segregation of oxygen. In conventional polycrystalline alloys, oxygen has been considered a detrimental element at GBs, contributing to brittleness due to its propensity to increase O-M (oxygen-metal) bonds while reducing M-M (metal-metal) bonds[9]. However, in the case of extreme nanocrystals with grain sizes around 10 nm, the critical nucleation stress for dislocations increases with decreasing grain size. In our study of the supra-nanograined CoCrNi-O MPEA, oxygen—a known embrittlement element in conventional polycrystalline materials—plays a significant role in enhancing the GB strength through massive O-M bonding. In addition, based on traditional interface thermodynamics, the segregation of oxygen at GBs can also reduce the excess energy of the GBs and stabilize them[77]. This is analogous to previous findings, where segregation of elements such as Mo in Ni has been shown to improve GB stability[13,78]. By increasing both the mechanical and thermal stability of GBs, the nucleation and propagation of shear bands and cracks at GBs is effectively inhibited. In particular, as deformation progresses, oxygen atoms tend to migrate along with gliding dislocations and become concentrated at GBs. This accumulation of oxygen results in a continuous increase in GB strength during deformation and subsequently provides continuous strain hardening of the material.

To convey a better understanding of the underlying mechanism of the oxygen redistribution dynamics and its impact on the superior mechanical properties, a schematic illustration is presented in Fig. 6. The stable plastic deformation of the CoCrNi-O MPEAs is carried by the high density of partial dislocations in the as-deposited sample, including Shockley partial dislocations gliding along the slip planes and Frank partial dislocations climbing along coherent twin boundaries. The formation of SFs and NTs induced by partial dislocations will, in turn, hinder subsequent dislocation motion, contributing to strain hardening. In addition, the evenly distributed (Cr, O)-rich clusters in the as-deposited sample also exert pinning of dislocations, creating rugged landscapes for the dislocation movement, thereby promoting a high yield strength and continuous strain hardening of the alloy. By the interaction with the dislocations, oxygen interstitials will be progressively dragged to the GBs, and thus increasingly segregate at the GBs, continuously strengthening and stabilizing them. Thus, GB sliding and the initiation of microcracks at the GBs are effectively inhibited, promoting the observed large uniform plastic deformation of the CoCrNi-O ultrafine nanocrystalline alloys.

In conclusion, by deliberately introducing a large amount of interstitial oxygen atoms, the supra-nano grained CoCrNi-O MPEAs with uniformly distributed oxygen-rich clusters have been synthesized. In the model (CoCrNi)$_{87}$O$_{13}$ alloy, intra-grain (Cr, O)-rich clusters can retard dislocation motion and cause dislocations to accumulate in the grain interior. Simultaneously, the (Cr, O)-rich clusters gradually migrate towards the GBs during deformation, leading to enhanced GB

strength and stability. Consequently, the $(CoCrNi)_{87}O_{13}$ alloy with a grain size of only 3 nm achieves an ultra-high yield strength of ~3.6 GPa and a uniform compressive plastic strain of over 50%. Our findings demonstrate that the addition of interstitials effectively enhances the chemical inhomogeneity and facilitates GB segregation. This mechanism not only stabilizes GBs, but also significantly boosts dislocation multiplication and storage capabilities within nanocrystalline grains. Our approach provides a solution to synergistically improve both the mechanical stability and strength of nanocrystalline metals, paving the way for the design of outstanding materials.

## Methods

### Materials preparation
Using Co, Cr, Ni, and NiO with a purity greater than 99.95% and a particle size of 30 μm as raw materials, a powder mixture was prepared in a molar ratio of 3:3:3:1 for Cr, Co, Ni, and NiO, respectively. For comparison, a powder mixture with equimolar ratio of Cr, Co and Ni was also used. The mixture was ball-milled under a high-purity argon atmosphere and subsequently sintered through a hot press sintering furnace to obtain oxygen-containing CoCrNiO and pure CoCrNi pre-alloyed target materials. The oxygen content in the CoCrNiO and pure CoCrNi alloy targets was measured using ONH analyzer (LECO ONH 836), with oxygen content of 0.31 wt.% and 0.02 wt.%, respectively. CoCrNiO or CoCrNi films with a thickness of ~1-3 μm were deposited on Si (001) substrates by magnetron sputtering. Before deposition, the Si substrates were ultrasonically cleaned in ethanol and acetone and blown dry. The sputtering chamber was evacuated to a base pressure below $8 \times 10^{-4}$ Pa, and the Ar or Ar+$O_2$ pressure was maintained throughout the deposition process. The distance between substrates and target were 120 mm or 100 mm. During sputtering, the substrates rotated at a rate of 10 r/min without applying bias voltage. The sputtering direct current power was 60 W or 120 W. The detailed magnetron sputtering parameters are listed in Supplementary Tables 4 and 5. The phase diagrams of Co-O, Cr-O, and Ni-O systems were calculated using the Thermo-Calc TCOX10 database[79].

To verify the universality of the oxygen nanoclustering strategy, we extend our investigations to other alloy systems. $(FeCrNi)_{98}O_2$ (at.%) and $(HfNbTiZr)_{97}O_3$ (at.%) alloy targets were used for film sputtering at room temperature. The distance between substrates and targets was 120 mm. The background vacuum was $8 \times 10^{-4}$ Pa and the Ar working pressure was 0.30 Pa. The sputtering power was 60 W and the films with thickness of ~1.80 μm were deposited on Si (001) substrates.

### Structural characterization
Microstructures and phase analysis of the films were examined by using TEM (JEOL F200operated at 200 kV). STEM images and EELS spectra were taken on a spherical aberration-corrected STEM (JEOL JEM-ARM300F, operated at 300 kV) equipped with a Gatan GIF Continuum K3 system. The spectrometer was configured with an energy dispersion of 0.18 eV/channel, and the collection and convergence angles were 24.75 mrad and 40.99 mrad, respectively. The TEM samples were prepared using a focused ion beam (FIB) workstation (Helios NanoLab 600i). The final milling was performed at 2 kV and 23 pA, ensuring minimal FIB-induced damage.

### Elemental composition analysis at the atomic scale
Atom probe tomography (APT) (CAMECA LEAP 4000X HR) was employed to detail the elemental compositions of the CoCrNi-O MPEAs. The specimens were analyzed under laser mode at a temperature of 50 K, with a pulse energy of 60 pJ, an evaporation detection rate of 1% atom per pulse, and a pulse rate of 200 kHz. Data analysis and three-dimensional (3D) atom map reconstruction were performed using the CAMECA IVAS 3.8 software. The APT samples were fabricated using annular milling in a FIB workstation, wherein the current was

progressively reduced as the specimen diameter decreased. The final parameters used for cleaning were 2 kV/23 pA.

### Chemical bonding behavior analysis
The chemical bonding behavior of the alloy was characterized by XPS and EELS spectra. The XPS spectra were acquired with a ThermoFisher ESCALAB 250Xi spectrometer equipped with a monochromatic Al Kα X-ray source. During argon ion etching, the ion energy was 2 keV and the etching time was 400 s. The white line method[80] was used to specifically analyze the electronic states of the CoCrNi-O MPEA based on recorded EELS spectra. Zero loss peak (ZLP) calibration was performed on each spectrum, followed by background subtraction using a power law function. In order to mitigate the effect of extended energy-loss fine structure (EXELFS) oscillations of the preceding element edge, the background subtraction windows were set at 12.60 eV away from the $L_3$ peak maximum of each element, with a window width of 5.40 eV. Due to the sample thickness being only 33 nm, which is much less than the mean free path of the inelastic scattering process at 300-keV incident energy, the influence of multiple scattering was limited[81]. Therefore, in the subsequent statistical analysis, the spectra were not deconvoluted to avoid the introduction of artifacts from this method. During the quantification analysis of the elemental concentration, the energy loss near edge structure (ELNES) of each element was excluded to improve the accuracy. In the white line intensity ratio statistics, the background intensity under the $L_2$ edge, arising from the continuum contributions of the $L_3$ edge, was subtracted using the Hartree-Slater mode function[82]. The function was fitted to a 1.98 eV-wide window that started 13.14 eV above the $L_2$ peak maximum, corresponding to the position at a relative minimum between $L_2$ and the first EXELFS oscillation. The integration interval was centered at the peak maximum, with a width of 4.00 eV for the Cr-L edge and 3.50 eV for the Co-L edge, respectively.

### Mechanical property characterization
Nanoindentation tests were performed using a Berkovich diamond indenter with a tip radius of ~100 nm on a commercial nanoindentation system (Hysitron TI 980 TriboIndenter, Bruker) at room temperature. The tip area function for the Berkovich tip was carefully calibrated on a fused silica standard sample before running the tests. The thermal drift was maintained at 0.05 nm/s for all experiments. The experiments were carried out using continuous stiffness measurement (CSM) in load-controlled mode with a peak load of 10 mN. The applied strain rate was set to be $0.005\,s^{-1}$, $0.01\,s^{-1}$, $0.05\,s^{-1}$ and $0.1\,s^{-1}$. The strain rate ($\dot{\varepsilon}$) is defined as $\dot{\varepsilon} = \frac{\dot{P}}{2P}$, where $P$ is the load and $\dot{P} = dP/dt$ is the loading rate[61]. The nanoindentation data were collected within a maximum indentation depth of 10–15% of the film thickness[83], and 9 indents were performed on each sample to obtain the average hardness ($H$) and reduced modulus ($E_r$). The Poisson's ratio for the CoCrNi-O film used for $E_r$ calculations is 0.30. The strain rate sensitivity is expressed as $m = \frac{\partial \ln H}{\partial \ln \dot{\varepsilon}}$, where $H$ is the hardness and $\dot{\varepsilon}$ is the applied strain rate. The apparent activation volume $V$ in indentation test can be calculated as $V = 3\sqrt{3}kT(\frac{\partial \ln \dot{\varepsilon}}{\partial H})$[60,61], where $k$ is the Boltzmann constant and $T$ is the absolute temperature ($T = 298$ K).

The compressive performance of the films was measured using a Hysitron TI 980 TriboIndenter (Bruker) by employing a flat diamond tip with a diameter of 10 μm at room temperature. The tests were conducted in feedback-enabled displacement-controlled mode at a specified nominal strain rate of $4 \times 10^{-3}\,s^{-1}$. To enhance the reliability of the experimental results, three repeated tests were conducted for each diameter micropillars under the same conditions. The micropillars for compression testing were fabricated in a FIB workstation, with precise control to maintain a 2:1 aspect ratio (height/width) and a taper angle smaller than 2°, minimizing the impact of pillar geometry (Supplementary Fig. 12). The stress was calculated by: $4P/[\pi(D_0 + 2l\tan\theta)^2]$[84], where $P$ is the load, $D_0$ is the tip diameter, $l$ is the tip displacement, and

$\theta$ is the taper angle. The strain was calculated from the ratio of the recorded displacement data to the height of the unloaded pillars after taking into account the machine stiffness[85]. The morphology of the pillars after deformation was further examined through the secondary electron imaging mode and backscattered electron imaging mode of an SEM (ZEISS Gemini 300).

## Critical grain size calculation

Based on the dislocation-based model proposed by Chen et al.[3], there is a critical grain size $d_c$, below which the critical shear stress needed to nucleate a partial dislocation will be less than for a perfect dislocation. The critical grain size can be expressed as

$$d_c = \frac{2\alpha G(b_N - b_P)b_P}{\gamma}, \quad (1)$$

where $G$ is the shear modulus ($G = \frac{E}{2(1+\nu)}$, $\nu$ is Poisson's ratio of 0.30), $\gamma$ is the SFE, $\alpha$ is a parameter related to the dislocation configuration as well as the grain size ($\alpha = 0.5$ and $1.5$ for edge and screw dislocations, respectively) and $b_N$ and $b_P$ are the magnitudes of the Burgers vectors of the perfect and the Shockley partial dislocation, respectively. Taking O-13 alloy as an example, $G = 63.98$ GPa, $\alpha = 1$, $b_N = a/2|<110>| = 2.49$ Å, $b_P = a/6|<211>| = 1.44$ Å ($a = 3.52$ Å), $\gamma = 22$ mJ·m$^{-2}$, $d_c$ was calculated as 87.94 nm for the O-13 MPEA. In addition, according to the model proposed by Nieh and Wadsworth[76], the critical size $l_c$ for a grain to accommodate single Shockley partial dislocation is given by

$$l_c = \frac{3Gb}{\pi(1-\nu)H}. \quad (2)$$

Taking O-13 alloy as an example, $G = 63.98$ GPa, $b = a/6|<211>| = 1.44$ Å, $\nu = 0.30$ and $H = 10.98$ GPa, $l_c$ was estimated to be 1.14 nm for the O-13 MPEA.

## Data availability

Source data are provided with this paper. All data supporting the findings of this study are available within the article and its Supplementary Information or from the corresponding authors upon request. Source data are provided with this paper.

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

## Acknowledgements

The research was supported by the National Natural Science Foundation of China (52171159), the Natural Science Foundation of Shanghai (22ZR1422500), the Shanghai Science and Technology Innovation Action Plan (No. 24CL2901500), the Technology Plan Program of Shanghai Municipal Commission of Science and Technology (No. 25CL2902300), the Shanghai Municipal Explorer Program (No. 25TS1401900), the Innovation Program of Shanghai Science and Technology (No. 23520760700), the Aviation Foundation (No. 2023Z0530S6004), and the Space Utilization System of China Manned Space Engineering (No. KJZ-YY-NCL08). C. Liu acknowledges the support from the open research fund of Suzhou Laboratory (No. SZLAB-1108-2024-TS001). We thank Dr. Jie Li at Shanghai University for the help with the nanoindentation and micro-compression experiments, and Dr. Weisen Zheng at Shanghai University for the help in phase diagram calculations.

## Author contributions

X.L.B. and G.W. (Shanghai University) proposed and supervised the research. X.L.B. and X.L.Y. designed the magnetron sputtering experiment. X.L.Y. carried out materials fabrication, mechanical, and microstructure characterizations. Y.F.L. and X.L.Y. performed TEM and STEM characterizations, and processed and interpreted the data under the guidance of C.L. and Q.W. X.L.Y. conducted an APT investigation and processed and interpreted the data with assistance from C.L., G.W. (Xi'an Jiaotong University), and Y.W. D.K. helped in nanoindentation and micro-compression analysis. X.L.Y., X.L.B., C.L., Q.W. and G.W. (Shanghai University) wrote and revised the manuscript. All authors, including Y.Y., D.Ş., and J.E., contributed to the discussion of the results and commented on the manuscript.

## Competing interests

The authors declare no competing interests.
