## [Transparent Peer Review file · Nature Communications]

Oxygen nanoclustering evades inverse Hall-Petch softening

Corresponding Author: Dr Xilei Bian

Version 0:

Reviewer comments:

Reviewer #1

(Remarks to the Author)

This is a piece of quite comprehensive work. However, it has several major flaws.

1. The wording of ductility was wrongly used in some places, including in the abstract where it was stated that a strategy to achieve superior strength and ductility was presented. They also stated something like this: impressively overcoming the strength ductility trade-off and the brittleness associated with extreme grain refinement. These statements are way too misleading: ductility is about tension, and it is not even relevant here.
2. They shall really make it obvious already in the abstract that strength and compressive strain are from micropillar tests, since otherwise it is misleading again.
3. Surprisingly, not even a single evident image of the so-called ultrafine columnar nanocrystalline structure was given. This is one thing. Another thing is that: how was the grain size defined? The width of the columnar structure? How about the length of the columnar structure? With an ill-defined grain size, it is not straightforward to discuss the grain size effect.
4. Their interpretation of Fig.4 is also strange, in the opinion of the reviewer. With only two data points being provided for CoCrNi-O, a conclusion on evading the Hall-Petch softening could not be drawn convincingly. Such an evasion was only seen for NiMo and NiCo. Not even for CoCrNi, since no data was provided for them at grain size smaller than 10 nm; one can certainly not refer to those data for doped CoCrNi alloys. They are not even the same materials. Therefore, the reviewer challenges the title of this work.
5. In the end, what is the point of developing these strong and malleable thin films?

Reviewer #2

(Remarks to the Author)

This study by Yu et al. presents an interesting and intriguing finding of oxygen nano-clustering as a means to evade Hall-Petch softening when grain size is reduced to the supranano scale (<10 nm). However, key issues regarding experimental control, composition selection, mechanistic interpretation, and comparisons with prior literature need to be addressed. Clarifying these aspects would significantly strengthen the validity and impact of the findings. Specific comments are shown as below.

1. Validity of the Conclusion on Hall-Petch Softening Evasion

The authors claim that oxygen nano-clustering at grain boundaries stabilizes the grain structure, preventing Hall-Petch softening in CoCrNi-O alloys when the grain size decreases below 10 nm. However, this conclusion is based on only two data points, where multiple variables—such as grain size and pre-deformation—are not independently controlled. To establish a solid conclusion, a controlled experiment isolating a single variable is necessary.

Additionally, the 2.5 nm sample was obtained from a pre-deformed alloy, which likely contains high-density dislocations and residual stress, both of which could influence the yield strength and plastic deformation behavior. A more convincing approach would involve using direct CoCrNi-O alloy samples with systematically varied grain sizes across the Hall-Petch softening threshold (e.g., from 100 nm down to 3 nm).

Furthermore, the mechanical behavior of CoCrNi alloys with grain sizes below 10 nm should be explicitly compared. Specifically, a CoCrNi alloy exhibiting Hall-Petch softening at this grain size should be included, followed by a comparison with oxygen-nanoclustered CoCrNi-O alloys to confirm the suppression of softening. As shown in Fig. 4, the CoCrNi alloy with a 26 nm grain size already exhibits a yield strength of 4.2 ± 0.3 GPa (J. Mater. Sci. Tech., 2022, 106, 1-9), which surpasses that of the CoCrNi-O alloy with a 2.5 nm grain size (4.16 ± 0.10 GPa). This suggests that the yield strength at the threshold grain size could be even higher than 4.2 GPa. If so, Hall-Petch softening may still occur, as the CoCrNi-O alloy with a 3 nm grain size has a lower yield strength of 3.6 GPa. Comparing this trend with Hall-Petch behavior in CoCrNi alloys doped with different elements is not straightforward due to differences in chemical composition. To substantiate the

proposed mechanism, the study should provide data where grain size is the sole controlled variable.

2. Dislocation Accumulation and Pre-Deformation Effects

The authors propose that O-rich clusters within grain interiors promote dislocation accumulation and multiplication. However, it is unclear how dislocations formed during pre-deformation are distinguished from those generated by the proposed mechanism. Clarification on this distinction is necessary to confirm the role of oxygen nano-clusters in influencing dislocation behavior.

3. Justification of Oxygen Composition and Generalizability

The rationale for selecting an oxygen content of 13 at.% remains unclear. What is the basis for this specific composition? How would variations in oxygen content influence mechanical properties? Additionally, further evidence is required to support the claim that this mechanism can be extended to other alloy systems beyond CoCrNi. Addressing these aspects would strengthen the broader applicability of the findings.

4. Comparison with Existing Literature

While the authors propose a novel strategy for enhancing strength and ductility via oxygen incorporation, similar approaches using interstitial oxygen solid solutions have been previously reported (Nat. Commun., 2022, 13, 1102). Moreover, the strength-strain comparison in Figure 3 overlooks several high-performance results from the literature, including yield strengths of 4.1 GPa with 50% uniform strain (Adv. Mater., 2020, 32, 2002619) and 4.0 GPa with 30% uniform strain (Nano Lett., 2025, 25, 691–698). A more comprehensive comparison with prior studies is necessary to contextualize the contribution of this work.

Furthermore, the claim of achieving over 50% uniform strain requires clarification, as the presented stress-strain curves only extend to 50%. Lastly, Figure 3 contains a labeling error—panel (f) is missing from the figure caption and should be corrected.

5. Oxidation State Analysis

In Figure 2, Cr is shown to exist in significant positive valence states in addition to the 0 valence state, suggesting notable oxidation. How do the authors confirm that these oxidation states arise solely from electron extraction by oxygen rather than oxide formation? (Page 5, Lines 7–9). Additionally, some Co peaks in Figure 2 are not fitted, which should be addressed to ensure accurate characterization of the oxidation state.

6. Dislocation Density in Fine-Grained Alloys

The generation of high-density dislocations at such fine grain sizes is generally difficult. Is this effect primarily driven by the low stacking fault energy of CoCrNi alloys, or does the substantial interstitial oxygen content play a more dominant role? Furthermore, can this mechanism be effectively controlled or tuned to optimize mechanical performance? Additional discussion and supporting evidence on this point would enhance the clarity of the key finding.

Version 1:

Reviewer comments:

Reviewer #1

(Remarks to the Author)

The reviewer is satisfied with the responses from the authors, and the revised manuscript can now be accepted.

Reviewer #2

(Remarks to the Author)

The authors have addressed the reviewers' comments with appropriate corrections and revisions, and the overall quality of the manuscript has been significantly improved. However, a few minor issues should be clarified before publication:

(1) In the abstract, the sentence "alloy exhibits a remarkable yield strength of ~3.6 GPa and retains a uniform deformation of over 50% strain under micropillar compression" reads somewhat awkwardly. It could be refined to better emphasize the combined mechanical performance.

(2) In Supplementary Fig. 14, the composition of FeCrNi-O should be specified.

(3) The number of significant digits after the decimal point in Supplementary Table 4 is inconsistent. Please also check for consistency throughout the manuscript.

(4) In Fig. 4a, the slope of the O-free alloy in the elastic region appears smaller than that of the other samples. The authors should provide an explanation for this observation.

(5) Please verify the y-axis labeling in Supplementary Fig. 2c.

Response to the Reviewers' Report

NCOMMS-24-86057

To Reviewer #1

General Comments: *“This is a piece of quite comprehensive work. However, it has several major flaws”*

Response: We are very grateful for the positive comments and support for our work. We have carefully addressed all the concerns raised by the reviewer point-to-point as follows. We believe that the quality of the manuscript has been significantly improved in the revised version. We hope the revised manuscript can fulfill your high requirements for the publication in *Nature Communications*.

Question 1: *“The wording of ductility was wrongly used in some places, including in the abstract where it was stated that a strategy to achieve superior strength and ductility was presented. They also stated something like this: impressively overcoming the strength ductility trade-off and the brittleness associated with extreme grain refinement. These statements are way too misleading: ductility is about tension, and it is not even relevant here.”*

Response: We thank the reviewer for pointing out our wrong phrasing. We have replaced “ductility” by “plasticity” in revised manuscript.

Question 2: *“They shall really make it obvious already in the abstract that strength and compressive strain are from micropillar tests, since otherwise it is misleading again.”*

Response: Thanks for your valuable suggestion. We have revised this sentence to declare that the strength and uniform strain are from the micropillar compression.

Question 3: *“Surprisingly, not even a single evident image of the so-called ultrafine columnar nanocrystalline structure was given. This is one thing. Another thing is that: how was the grain size defined? The width of the columnar structure? How about the length of the columnar structure? With an ill-defined grain size, it is not straightforward to discuss the grain size effect.”*

Response: Thank you very much for the comments.

In our original manuscript, we have shown a DF-TEM image of the (CoCrNi)₈₇O₁₃ (O-13) alloy before deformation in **Supplementary Fig. 9** (now **Supplementary Fig. 16, revised version**) and three high-resolution TEM images of the columnar grains in **Supplementary Fig. 4** (now **Supplementary Fig. 8, revised version**), both of which can prove that the as-deposited O-13 alloy has an ultrafine columnar nanocrystalline structure. In addition, in our revised Supplementary Materials, the BF-STEM images in **Supplementary Fig. 5** and DF-TEM images in **Supplementary Fig. 6** clearly reveal that the pure CoCrNi, (CoCrNi)₉₅O₅ (O-5) and (CoCrNi)₇₀O₃₀ (O-30) have similar

ultrafine columnar nanocrystalline structure.

For the definition of grain size, we have explicitly detailed in the revised manuscript that the grain size (d) is defined by the width of the columnar grains. Please see Lines 4-9 on Page 4 in the revised manuscript.

Taking O-13 alloy as an example, the average length (l_{average}) of the columnar structure is statistically measured as 10 nm (Fig. R1), which is more than triple the width. Due to the extremely small columnar grain width, dislocations in the grain interior could rapidly run into the columnar grain boundary, and the dislocation motion are obstructed. Thus, it is the very small width of the columnar grain that substantially reduce the mean free path of dislocations. Since Hall-Petch effect primarily arises from the obstruction of dislocation motion by grain boundaries (GBs), it is reasonable to take the width of columnar grains as a measurement for grain size. This definition has been widely accepted for columnar grained alloys [Nat. Commun. 13, 1102 (2022); Nat. Commun. 6, 7748 (2015); Sci Adv 7, eabg5113 (2021).]. We thus conclude that the CoCrNi, O-5, O-13, and O-30 alloys show comparable grain sizes (columnar widths).

Fig. R1 The distribution of columnar length of O-13 MPEA.

Question 4: “Their interpretation of Fig.4 is also strange, in the opinion of the reviewer. With only two data points being provided for CoCrNi-O, a conclusion on evading the Hall-Petch softening could not be drawn convincingly. Such an evasion was only seen for NiMo and NiCo. Not even for CoCrNi, since no data was provided for them at grain size smaller than 10 nm; one can certainly not refer to those data for doped CoCrNi alloys. They are not even the same materials. Therefore, the reviewer challenges the title of this work.”

Response: Thank you very much for your constructable suggestions. We have to admit that the limited data points for CoCrNi-O and doped CoCrNi alloys cannot rigorously demonstrate the inverse Hall-Petch (H-P) softening behavior. To this end, we carried

out supplementary experiments including ten groups of O-free CoCrNi and two groups of CoCrNi-O MPEAs. It is found that the grain sizes of all current investigated alloys are well below 20 nm. The oxygen content of the CoCrNi-O MPEA is 4.81 at.% (O-5) and 30.25 at.% (O-30), respectively. The detailed grain size and deposition parameters are provided in Supplementary Tables 3 and 4. The composition analysis of O-5 and O-30 are shown in Supplementary Figs. 1 and 2. The structure characterizations of these investigated samples are shown in Supplementary Figs. 5-7.

For illustrating the inverse H-P effect in CoCrNi MPEA and the evasion of this effect, we measured the hardnesses of these samples by using nanoindentation tests. Figure R2 (also see Fig. 3 in the revised manuscript) summarizes the hardness vs. grain size data from both the literature and the present work, providing conclusive evidence for the coexistence of the H-P effect and inverse H-P effect in CoCrNi alloys. That is to say, below the threshold value (~ 15 nm), CoCrNi alloys show a breakdown of H-P relation, resulting in an inverse H-P relation, i.e., further reducing grain size leads to a decrease in strength. However, it is worth noting that the hardness values of CoCrNi-O alloys with varying oxygen contents (4.81–30.25 at.%) are much higher than the O-free CoCrNi alloys with similar grain sizes, lying above the fitted inverse H-P line. This clearly demonstrates that the introduction of oxygen nano-clusters can significantly evade inverse H-P softening in CoCrNi alloy.

We note that our analysis of the H-P and inverse H-P effects in these samples is based on hardness value. We further performed micropillar compression tests on CoCrNi-O MPEAs to demonstrate that our oxygen nano-clustering strategy not only effectively evades inverse H-P softening but also significantly enhances the uniform plastic deformability of nanocrystals.

We have replotted the figure and revised the related text. Please see Fig. 3 on Page 23 and the 3rd Paragraph on Page 5 in the revised manuscript.

Fig. R2 Evading inverse Hall-Petch softening in the CoCrNi alloy. Hardness versus average grain size for our current CoCrNi and CoCrNi-O alloys. Literature data of Ni⁴²⁻⁴⁵, NT-Ni⁴⁶, NiMo¹³, NiCo⁴⁷, and CoCrNi⁴⁸⁻⁵⁷ alloys are included for comparison. The solid lines are fitted by Hall-Petch relation ($H \sim d^{1/2}$). The dash lines show the inverse Hall-Petch softening ($H \sim d$).

Question 5: *“In the end, what is the point of developing these strong and malleable thin films?”*

Response: Thank you very much for the insightful comment. From a fundamental perspective, study on the strong and malleable thin films can deepen our understanding of the structure and mechanical properties of ultrafine-grained alloys, especially for exploring a general strengthening and toughening strategy for multicomponent alloys with grain sizes situated in the inverse Hall-Petch regime. With further investigations, our oxygen nano-clustering strategy indeed demonstrates broad applicability across alloy systems, including crystalline and amorphous system, for designing materials with unprecedented strength-deformability synergy (see **Response to Question 3 of Reviewer #2**). In terms of practical applications, these strong and malleable thin films show great potential in extreme mechanical environments requiring supreme wear resistance such as next-generation cutting tools and protective coatings. Additionally, two-dimensional (2D) metallic materials are compelling candidates for constructing three-dimensional (3D) micro- and nano-devices with complex architectures. (e.g., tubular, spiral, or reticular shapes) due to their unique mechanical and functional properties [Chem. Rev. 118, 6409-6455 (2018); Nature 639, 354-359 (2025)]. However, their high surface-to-volume ratio raises concerns about oxidation [Nat. Mater. 23, 52-57 (2024)]. Our findings on oxygen-induced strengthening and toughening in ultrafine nanocrystalline CoCrNi films offer critical insights into mitigating oxidation effects—particularly relevant for Co- and Ni-based 2D systems [Chem. Rev. 118, 6409-6455 (2018)]. We have highlighted this point in our revised manuscript. Please refer to **Lines 33-35 on Page 7**.

To Reviewer #2:

General Comments: “This study by Yu et al. presents an interesting and intriguing finding of oxygen nano-clustering as a means to evade Hall-Petch softening when grain size is reduced to the supra-nano scale (<10 nm). However, key issues regarding experimental control, composition selection, mechanistic interpretation, and comparisons with prior literature need to be addressed. Clarifying these aspects would significantly strengthen the validity and impact of the findings. Specific comments are shown as below.”

Response: Thank you very much for your precious support and appreciation of the novelty of our work. Your kind comments are very supportive to our present work and future work in this field. We have carefully addressed all the comments raised by the reviewer point-to-point as follows. We believe that the quality of the manuscript has

been significantly improved in the revised version.

“Validity of the Conclusion on Hall-Petch Softening Evasion”

Question 1.1: *The authors claim that oxygen nano-clustering at grain boundaries stabilizes the grain structure, preventing Hall-Petch softening in CoCrNi-O alloys when the grain size decreases below 10 nm. However, this conclusion is based on only two data points, where multiple variables—such as grain size and pre-deformation—are not independently controlled. To establish a solid conclusion, a controlled experiment isolating a single variable is necessary.*

Response: We are grateful to the reviewer’s helpful suggestion. In order to rigorously demonstrate the grain size effect on CoCrNi MPEA, we carried out supplementary experiments including ten groups of O-free CoCrNi and two groups of CoCrNi-O MPEAs by controlling the grain sizes well below 20 nm. The oxygen contents of the two groups of CoCrNi-O MPEAs are 4.81 at.% and 30.25 at.%; the two CoCrNi-O alloys are named as O-5 and O-30, respectively. Further characterization of mechanical properties of all samples clearly demonstrates that our oxygen nano-clustering strategy not only effectively evades inverse Hall-Petch (H-P) softening but also significantly enhances the uniform plastic deformability of CoCrNi MPEA. For more details, please see Response to Question 4 of Reviewer #1.

Question 1.2: *Additionally, the 2.5 nm sample was obtained from a pre-deformed alloy, which likely contains high-density dislocations and residual stress, both of which could influence the yield strength and plastic deformation behavior. A more convincing approach would involve using direct CoCrNi-O alloy samples with systematically varied grain sizes across the Hall-Petch softening threshold (e.g., from 100 nm down to 3 nm).*

Response: We thank the reviewer for the valuable comments and we are sorry for our unclear statement. Firstly, we would like to emphasize that, in our original manuscript, the Ext. 500 sample (i.e., 2.5 nm sample, after 30% strain deformation) is initially used to substantiate the presence of strain hardening rather than grain size effect of CoCrNi-O MPEA. Secondly, the grain size effect on CoCrNi-O MPEA is not the addressing issues in this work, we more focus on the grain size effect on pure CoCrNi alloy and the influence of the oxygen doping on the evading of the inverse Hall-Petch softening of such supra-nano grained CoCrNi alloys. Lastly, we attempted to adjust the parameters of the magnetron sputtering to prepare CoCrNi-O MPEAs with identical oxygen content while varying grain sizes. However, our experimental results reveal that altering any sputtering parameter will simultaneously affect the oxygen content and the grain size. As shown in Fig. 3 in the revised manuscript, our extra CoCrNi-O results show that the average grain sizes of O-5 and O-30 are 8.18 nm and 6.02 nm, respectively. The hardness of the two alloys (O-5 and O-30) is comparable to the O-13 alloy. That is to say, the oxygen content and the grain size jointly determine the mechanical properties. Therefore, preparing CoCrNi-O MPEAs with grain size as the sole variable requires extensive further experimentation. We will systematically carry out in-depth investigations on this issue in our future work.

Question 1.3: *Furthermore, the mechanical behavior of CoCrNi alloys with grain sizes below 10 nm should be explicitly compared. Specifically, a CoCrNi alloy exhibiting Hall-Petch softening at this grain size should be included, followed by a comparison with oxygen-nanoclustered CoCrNi-O alloys to confirm the suppression of softening. As shown in Fig. 4, the CoCrNi alloy with a 26 nm grain size already exhibits a yield strength of 4.2 ± 0.3 GPa (J. Mater. Sci. Tech., 2022, 106, 1-9), which surpasses that of the CoCrNi-O alloy with a 2.5 nm grain size (4.16 ± 0.10 GPa). This suggests that the yield strength at the threshold grain size could be even higher than 4.2 GPa. If so, Hall-Petch softening may still occur, as the CoCrNi-O alloy with a 3 nm grain size has a lower yield strength of 3.6 GPa. Comparing this trend with Hall-Petch behavior in CoCrNi alloys doped with different elements is not straightforward due to differences in chemical composition. To substantiate the proposed mechanism, the study should provide data where grain size is the sole controlled variable.*

Response: We appreciate the reviewer's valuable suggestion. With respect to the mechanical properties and inverse H-P effect in CoCrNi alloys with grain sizes below 10 nm, we incorporated 10 additional control groups of CoCrNi with grain sizes below 20 nm and performed nanoindentation tests. It is found that reducing grain size leads to a decrease in hardness, thereby confirming the presence of inverse H-P softening in CoCrNi alloys (see Fig. R2 or Fig. 3). Moreover, the two newly added CoCrNi-O alloy groups, as well as the original O-13 sample, substantiate the role of oxygen nano-clustering in evading inverse Hall-Petch softening in CoCrNi alloys. Please refer to our Response to Question 4 of Reviewer #1 for more details.

By comparing the hardness (14 GPa) and yield strength (4.2 GPa) of CoCrNi with a grain size of 26 nm [J. Mater. Sci. Tech., 106, 1-9 (2022)] to the strength and hardness values obtained of all samples in our work, one can note that the strength and hardness of CoCrNi and CoCrNi-O MPEAs with grain size below 20 nm are indeed lower (Fig. R3). This suggests that the grain size range for the transition from H-P strengthening to softening may lie between 17 nm and 26 nm. However, in comparison with the softening observed in pure CoCrNi (as fitted by the inverse H-P curve), it is clear that the hardness (strength) values of CoCrNi-O alloys are much higher than the O-free CoCrNi alloys with similar grain sizes, confirming the introduction of oxygen nano-clusters in CoCrNi alloy effectively mitigates GB-induced softening (Fig. R3). To highlight our findings, thus we didn't include this data point from J. Mater. Sci. Tech., 106, 1-9 (2022) in our revised manuscript (see Fig. 3).

Fig. R3 Evading inverse Hall-Petch softening in the CoCrNi-O MPEA. a Hardness (H) versus grain size and **b** yield strength ($H/3$ & σ_y) versus grain size for the CoCrNi and CoCrNi-O alloys. The solid lines are fitted by Hall-Petch relation. The dash lines show the inverse Hall-Petch softening.

“Dislocation Accumulation and Pre-Deformation Effects”

Question 2: *The authors propose that O-rich clusters within grain interiors promote dislocation accumulation and multiplication. However, it is unclear how dislocations formed during pre-deformation are distinguished from those generated by the proposed mechanism. Clarification on this distinction is necessary to confirm the role of oxygen nano-clusters in influencing dislocation behavior.*

Response: We thank the reviewer for the valuable comments. As we declared in **Response to Question 1.2 of Reviewer #2**, the pre-deformation sample is used to verify the presence of strain hardening in the $(\text{CoCrNi})_{87}\text{O}_{13}$ (O-13) sample, and is not directly involved in the discussion of grain size effects. It is noteworthy that the pre-deformation of O-13 sample (e.g. Ext. 500 nm) also contains oxygen nano-clusters. Therefore, our discussion here is limited to the role of O-rich clusters within grain interiors in influencing dislocation behavior. The oxygen nano-clusters significantly influences the dislocations accumulation and multiplication during deformation. For example, Lei et al. reported that oxygen-rich clusters can interact with moving dislocations and promote double cross-slip, which further enhances dislocation multiplication [Nature 563, 546-550 (2018)]. According to Song et al., interstitial oxygen atoms increase the local charge density, leading to additional bonding contributions [Nature 618, 63-68 (2023)]. This can decelerate the motion of surrounding dislocations, increases the probability of dislocation interactions [J. Mech. Phys. Solids 176 (2023)]. Thus, O-rich clusters within grain interiors plays a pivotal role in promoting dislocation multiplication [Nat. Commun. 13, 1102 (2022)]. For our current work, in situ observation of intragranular dislocation behavior during deformation is exceedingly challenging for grains smaller than 10 nm. Therefore, we employed a quasi-in situ approach by conducting TEM characterization of micropillars subjected to varying degrees of compressive strain to investigate the dislocation dynamics during deformation. In micropillars deformed to 50% strain, stacking faults and their associated partial dislocations within the grains can

be distinctly observed (see Fig. 5a-c). These findings demonstrate that, under the influence of O-rich clusters, dislocation reactions continue to play a critical role in the multiplication of partial dislocations in CoCrNi-O MPEA [Phys. Rev. Lett. 105, 135501 (2010)] and persist throughout the deformation process.

“Justification of Oxygen Composition and Generalizability”

Question 3: *The rationale for selecting an oxygen content of 13 at.% remains unclear. What is the basis for this specific composition? How would variations in oxygen content influence mechanical properties? Additionally, further evidence is required to support the claim that this mechanism can be extended to other alloy systems beyond CoCrNi. Addressing these aspects would strengthen the broader applicability of the findings.*

Response: We thank the reviewer for the suggestion. We acknowledge that the initial experimental scope in our manuscript was insufficient. The selection of 13 at.% oxygen content was primarily based on the observation of unique mechanical behavior in this sample prepared from oxygen-containing alloy targets. To elucidate how oxygen content influences the mechanical properties, we systematically modulated the oxygen solid-solution content in CoCrNi alloys. The revised manuscript now includes detailed characterization of CoCrNi-O alloys with oxygen content of 4.81 at.% (O-5) and 30.25 at.% (O-30), confirming that the O-13 MPEA (13 at.% O) has the smallest grain size and exhibits an optimal combination of strength and plasticity (see Fig. 4 in our revised manuscript). Please also see our Response to Question 4 of Reviewer #1.

Besides CoCrNi alloy system, in FeCrNi system, we also achieved similar oxygen doping strategy with oxygen content of 6.86 at.% ((FeCrNi)₉₃O₇) and the oxygen-induced chemical heterogeneity (Fig. R4a-b). The average grain size of the (FeCrNi)₉₃O₇ MPEA is 7.4 nm, which is already in the inverse Hall-Petch regime (Fig. R4c). However, compared with the low strength (hardness ~ 7.80 GPa) [J. Mater. Sci. 47, 1562-1566 (2011)], our FeCrNi-O MPEA achieves a high yield strength of 2.98 GPa (hardness ~ 9.00 GPa) and a large uniform strain of over 50% upon micropillar compression (Fig. R4d), confirming the strengthening-toughening effects mediated by oxygen-rich clustering in FeCrNi alloy system. In addition, our oxygen nano-clustering strategy can also extend to amorphous system. For example, amorphous HfNbTiZr-O thin films with an oxygen content of approximately 15.00 at.% were fabricated via magnetron sputtering. Micropillar compression results in Fig. R5 demonstrate a remarkable yield strength of 4.20 GPa, substantially surpassing the previously reported value of 1.80 GPa in the literature [Materials Today 51, 6-14 (2021)]. Therefore, we believe that our oxygen nano-clustering strategy can extend beyond specific compositions, demonstrating universal applicability across diverse alloy systems. We have highlighted this point. Please see the Second Paragraph, Page 7 in the revised manuscript.

Fig. R4 Composition, microstructure and mechanical properties of FeCrNi-O MPEA. **a** Individual 3D elemental maps comprised with specific element content. **b** 1D compositional profile along the arrows displayed in **a**. **c** HRTEM image along with the grain size distribution and corresponding FFT image. **d** Typical compressive stress-strain curve and corresponding SEM image of deformed micropillar.

Fig. R5 Microstructure and mechanical properties of amorphous (HfNbTiZr)₈₅O₁₅ MPEA. **a** XRD pattern and TEM image of the as-deposited film. **b** Typical compressive stress-strain curve and corresponding SEM image of deformed micropillar. The stress-strain curve of HfNbTiZr amorphous alloy is also plotted for comparison.

“Comparison with Existing Literature”

Question 4.1: *While the authors propose a novel strategy for enhancing strength and ductility via oxygen incorporation, similar approaches using interstitial oxygen solid solutions have been previously reported (Nat. Commun., 2022, 13, 1102). Moreover, the strength-strain comparison in Figure 3 overlooks several high-performance results from the literature, including yield strengths of 4.1 GPa with 50% uniform strain (Adv. Mater., 2020, 32, 2002619) and 4.0 GPa with 30% uniform strain (Nano Lett., 2025, 25, 691–698). A more comprehensive comparison with prior studies is necessary to contextualize the contribution of this work.*

Response: We thank the reviewer for the valuable suggestion. The first work [Nat. Commun. 13, 1102 (2022)] provided by the reviewer investigates oxygen addition in a body-centered cubic (BCC) TiZrNb alloy. In contrast, our work focuses on face-centered cubic (FCC) alloy. Compared with BCC alloys, FCC alloys typically exhibit higher plasticity but lower strength, and researchers often aim to enhance their strength without significantly compromising plasticity. Strategies such as grain refinement and interstitial solid solution strengthening are commonly employed to achieve this goal. Therefore, exploring interstitial solid solution strategies specifically for FCC nanocrystalline alloys is of great scientific significance. Moreover, our achieved oxygen concentration (30 at.%) far exceeds the values reported in the literature [Nat. Commun. 13, 1102 (2022)].

Regarding the scope of literature comparisons in original manuscript, we add clarification in the revised manuscript that all alloys compared in Fig. R6 (also see Fig. 4 in the revised manuscript) are exclusively FCC structured systems. To maintain consistency, we have removed the original comparisons with crystal-glass nanocomposites, as their distinct structural characteristics fall outside the focus of this study on FCC alloys. Therefore, we selectively compared only the micropillar compression results of nanocrystalline CoCrNi from the references provided by the reviewer. Specifically, the yield strength and uniform strain of CoCrNi are reported as 3.3 GPa and ~12% [Adv. Mater. 32, 2002619 (2020)] and 3.4 ± 0.1 GPa and 17% [Nano Lett. 25, 691–698 (2025)], respectively. We have added the two data points in Fig. R6 (Fig. 4 in the revised manuscript). Accordingly, we have revised related text. Please see Lines 13-15, Page 7 in the revised manuscript.

As for the high-performance results from the literature, including yield strengths of 4.1 GPa with 50% uniform strain [Adv. Mater. 32, 2002619 (2020)] and 4.0 GPa with 30% uniform strain [Nano Lett. 25, 691–698 (2025)], the mechanical properties are corresponded to crystalline-glass composites not nanocrystalline alloy. To highlight our findings and make reasonable comparison, we also prepared CrCoNi-O/TiZrNbHf-O crystalline-amorphous nanolayered composites via magnetron sputtering and investigated the mechanical properties via micropillar compression. As shown in Fig. R7, it is evident that our nanolayered material exhibits an ultra-high yield strength of ~4.20 GPa with a uniform strain of over 50%, which are comparable to the results of the above two studies [Adv. Mater. 32, 2002619 (2020); Nano Lett. 25, 691–698

(2025)], but far exceeding that of the same alloy systems (CrCoNi/TiZrNbHf) without oxygen doping reported as Wu et al. [Mater. Today 51, 6-14 (2021)]. This indicates that we can also achieve exceptional strength-deformability combination through fabrication of oxygen-doped composites. Again, these results are out of our current scope. Thus, we now only make comparisons of FCC structured alloy systems.

Fig. R6 Mechanical properties of the CoCrNi-O MPEAs. **a** Compressive engineering stress-strain curves of the micropillars with diameters of 500 nm and 1 μm of the CoCrNi-O alloys tested with identical conditions at room temperature. Insert shows how the Ext. 500 (extracted 500 nm) micropillars were obtained. **b-d** SEM images of the tested O-13 micropillars after compression. No shear bands can be observed. **e** Yield strength vs. uniform strain of the CoCrNi-O alloys, in comparison with other FCC structured alloy systems, including O-free CoCrNi alloy^{32,34,48,49,53,63-70}, pure Ni metal with grain size of 80 μm (CG) and 20 nm (NG-20) or twin thickness of 6.4 nm (NT-6.4) and 2.9 nm (NT-2.9)⁴⁶, AlFeCoNiC_x ($x=0, 0.5, 1.0, 2.0, 4.1$) alloy⁷¹, and Al_{0.1}CoCrFeNi alloy⁷² tested in micropillar compression at ambient temperature. The data of this work are located at upper-right corner.

Fig. R7 Comparison of the compressive engineering stress-strain curves of the CoCrNi/TiZrNbHf crystalline-amorphous nanolayered composites with and without oxygen. Inset shows the SEM morphology of the sample after ~ 50% strain deformation.

Question 4.2: “Furthermore, the claim of achieving over 50% uniform strain requires clarification, as the presented stress-strain curves only extend to 50%. Lastly, Figure 3 contains a labeling error—panel (f) is missing from the figure caption and should be corrected.”

Response: Thank you very much for your comments. We have supplemented the study with a micropillar compression test of O-13, which demonstrates the uniform strain can be up to 75%, validating our claim. We also have revised the wrong labeling. Please see the revised Fig. 4 and figure captions on Page 24.

“Oxidation State Analysis”

Question 5: In Figure 2, Cr is shown to exist in significant positive valence states in addition to the 0 valence state, suggesting notable oxidation. How do the authors confirm that these oxidation states arise solely from electron extraction by oxygen rather than oxide formation? (Page 5, Lines 7–9). Additionally, some Co peaks in Figure 2 are not fitted, which should be addressed to ensure accurate characterization of the oxidation state.

Response: We thank the reviewer for the valuable comments. Whether substitutional or interstitial oxygen atoms, both will lead to changes in the electronic states of metal atoms [J. Am. Chem. Soc. 128, 15666-15671 (2006); Phys. Rev. B 94 (2016)]. However, there is no evidence for oxides formation in the analysis of APT and TEM results. Moreover, the peak energy and the integrated L_3/L_2 intensity ratio of Cr and Co element based on the EELS data are between those of oxides and pure metals (Supplementary Fig. 9). Therefore, we believe that the oxidation state of the metal observed in the XPS results arises from the additional bond contribution induced by interstitial oxygen.

For the high-resolution XPS fitting of Co, we applied a multiplet peak fitting approach

that differs from the Gaussian fitting used for the other three elements. Multiplet peak fitting was employed exclusively for Co because the XPS spectra of the other three elements could be well-fitted using Gaussian functions, whereas the Co peaks in our experiments could not be satisfactorily fitted in this way, prompting us to further consider the effects of multiple splitting for Co [Appl. Surf. Sci. 257, 2717-2730 (2011)]. Currently, fitting parameter references for multiplet peak fitting approach are only available for the $2P_{3/2}$ peak of Co, so we did not attempt to fit the $2P_{1/2}$ peak without proper literature guidance.

“Dislocation Density in Fine-Grained Alloys”

Question 6.1: *The generation of high-density dislocations at such fine grain sizes is generally difficult. Is this effect primarily driven by the low stacking fault energy of CoCrNi alloys, or does the substantial interstitial oxygen content play a more dominant role?*

Response: We thank the reviewer for the comment. The low stacking fault energy (SFE) of CoCrNi alloys plays a significant role in the generation of dislocations. This is because the SFE determines the critical nucleation stress of partial dislocations [Materials Today 85, 17-26 (2025); Nat. Commun. 8, 14390 (2017)]. Taking a SFE of $22 \text{ mJ}\cdot\text{m}^{-2}$, the critical grain size of CoCrNi alloy is calculated to be approximately 87.94 nm (see Methods), which is significantly larger than the grain size in the inverse Hall-Petch regime ($<17\text{-}26 \text{ nm}$). The CoCrNi-(O) alloys can easily overcome the energy barrier and continuously form partial dislocations under the action of external loads. Thus, the high density of intra-granular partial dislocations in the samples before and after deformation is primarily attributed to the extremely low SFE ($\sim 22 \text{ mJ}\cdot\text{m}^{-2}$) of the CoCrNi alloy. We have detailed this point in the **First and Second Paragraphs, Discussion** in our manuscript

In addition, based on the nanoindentation results, we find that all investigated CoCrNi-O alloys (O-5, O-13, and O-30) exhibit a comparable activation volume ($\sim 30b^3$). This indicates that the oxygen content has marginal effect on the density of dislocations. Furthermore, many investigations have demonstrated that interstitial oxygen atoms provide additional bonding contributions and reduce the mobility of surrounding dislocations, thereby increasing the probability of dislocation reactions and promoting dislocation multiplication [Nature 563, 546-550 (2018); Nature 618, 63-68 (2023); Nat. Commun. 13, 1102 (2022)]. This effect is observed in both bulk and nanocrystalline samples. In the case of CoCrNi alloys, which inherently have low SFE. The low SFE decreases the nucleation stress for partial dislocations, and the obstructions for dislocation motion reduce the mean free path of partial dislocations. Thus, upon deformation, the increased tendency of dislocation interactions induced by massive incorporation of interstitial oxygen atoms can significantly enhance the proliferation of partial dislocations.

Question 6.2: *Furthermore, can this mechanism be effectively controlled or tuned to optimize mechanical performance? Additional discussion and supporting evidence on this point would enhance the clarity of the key finding.”*

Response: We are grateful for the reviewer's comments. Altering the oxygen content has been proven to effectively tune the mechanical properties of the CoCrNi alloy (see Figs. 3 and 4 in the revised manuscript). As seen in Fig. 4, among the current investigated three O-containing CoCrNi (O-5, O-13, and O-30) alloys, the O-13 alloy has the smallest grain size and exhibits an optimal combination of strength and plasticity. More detailed discussion can be found in the Last Paragraph on Page 6 and the First Paragraph on Page 7.

Currently, there are two primary methods for introducing oxygen into deposited samples: (1) using target materials that inherently contain oxygen, and (2) introducing a mixed atmosphere of oxygen and argon during the sputtering process. Additionally, adjusting sputtering parameters—such as sputtering power, target-to-substrate distance, sputtering pressure, and target angle—can further modulate the oxygen content in the deposited samples. However, since grain size is also influenced by deposition parameters, extensive orthogonal experiments are required to establish precise control due to the complexity of these interdependent variables.

Response to the Reviewers' Report

NCOMMS-24-86057A-Z

To Reviewer #1

General Comments: *“The reviewer is satisfied with the responses from the authors, and the revised manuscript can now be accepted.”*

Response: We are very grateful to the reviewer for the supportive comments.

To Reviewer #2:

General Comments: *“The authors have addressed the reviewers' comments with appropriate corrections and revisions, and the overall quality of the manuscript has been significantly improved. However, a few minor issues should be clarified before publication.”*

Response: We are very grateful to your recognition of our work. We have carefully addressed all the comments raised by the reviewer, as shown below in detail. Thanks to the reviewer's valuable suggestions, we believe that the quality of the manuscript has been significantly improved in the revised version.

Question 1: *“In the abstract, the sentence “alloy exhibits a remarkable yield strength of ~3.6 GPa and retains a uniform deformation of over 50% strain under micropillar compression” reads somewhat awkwardly. It could be refined to better emphasize the combined mechanical performance.”*

Response: Thanks for your valuable suggestion. This sentence has been revised as: alloy exhibits an exceptional yield strength of ~3.6 GPa and retains a uniform plastic strain of over 50% under micropillar compression. Please see the revised **Abstract**.

Question 2: *“In Supplementary Fig. 14, the composition of FeCrNi-O should be specified.”*

Response: We are grateful for your comment. The composition of FeCrNi-O has been specified by $(\text{FeCrNi})_{93}\text{O}_7$ in **Supplementary Fig. 14**.

Question 3: *“The number of significant digits after the decimal point in Supplementary Table 4 is inconsistent. Please also check for consistency throughout the manuscript.”*

Response: Thanks for the reviewer's insightful comment. The number of significant digits after the decimal point is kept for two. Please see the **revised Supplementary Table 4**. Besides, we have carefully checked the manuscript thoroughly for consistency. Please see these numbers marked by red color.

Question 4: *“In Fig. 4a, the slope of the O-free alloy in the elastic region appears smaller than that of the other samples. The authors should provide an explanation for*

this observation.”

Response: We thank the reviewer for this insightful comment. In Fig. 4a, the slope in the elastic region corresponds to Young's modulus, where a smaller slope indicates a lower Young's modulus that caused by weaker atomic bonding. This is reasonable for the O-free alloy, as metal-oxygen bonds (ionic/covalent) are typically stronger than metal-metal bonds. Thus, the absence of oxygen results in a lower overall bond strength and a smaller slope.

Question 5: *“Please verify the y-axis labeling in Supplementary Fig. 2c.”*

Response: We thank the reviewer for the valuable comment. The y-axis labeling in **Supplementary Fig. 2c** has been changed to "Bulk normalized concentration".